# Smartphone LiDAR Data: A Case Study for Numerisation of Indoor Buildings in Railway Stations

**DOI:** 10.3390/s23041967

**Published:** 2023-02-09

**Authors:** Orphé Catharia, Franck Richard, Henri Vignoles, Philippe Véron, Améziane Aoussat, Frédéric Segonds

**Affiliations:** 1SNCF Gares & Connexions, 75013 Paris, France; 2LCPI, Arts et Métiers Institute of Technology, HESAM Université, 75013 Paris, France; 3LISPEN, Arts et Métiers Institute of Technology, HESAM Université, 13617 Aix-en-Provence, France

**Keywords:** LiDAR smartphone, BIM, 3D laser scanner, 3D point cloud, spatial data analysis, digital mockup

## Abstract

The combination of LiDAR with other technologies for numerisation is increasingly applied in the field of building, design, and geoscience, as it often brings time and cost advantages in 3D data survey processes. In this paper, the reconstruction of 3D point cloud datasets is studied, through an experimental protocol evaluation of new LiDAR sensors on smartphones. To evaluate and analyse the 3D point cloud datasets, different experimental conditions are considered depending on the acquisition mode and the type of object or surface being scanned. The conditions allowing us to obtain the most accurate data are identified and used to propose which acquisition protocol to use. This protocol seems to be the most adapted when using these LiDAR sensors to digitise complex interior buildings such as railway stations. This paper aims to propose: (i) a methodology to suggest the adaptation of an experimental protocol based on factors (distance, luminosity, surface, time, and incidence) to assess the precision and accuracy of the smartphone LiDAR sensor in a controlled environment; (ii) a comparison, both qualitative and quantitative, of smartphone LiDAR data with other traditional 3D scanner alternatives (Faro X130, VLX, and Vz400i) while considering three representative building interior environments; and (iii) a discussion of the results obtained in a controlled and a field environment, making it possible to propose recommendations for the use of the LiDAR smartphone at the end of the numerisation of the interior space of a building.

## 1. Introduction

The 3D numerisation of the environment using new handheld sensors is a rather vast subject around which new forms of technology have emerged such as smartphone LiDAR (Light Detection And Ranging) and handheld scanners. Despite the many difficulties that these sensors face today, particularly in terms of the accuracy and quality of the data returned, they are nevertheless new tools that seem more suitable in terms of use, particularly in the numerisation of buildings.

Most research dealing with interior modelling or, more recently, the creation of as-built Building Information Modelling (BIM) [1] models, uses Terrestrial Laser Scanners (TLS) to collect datasets in the form of point clouds before converting them to Computer-Aided Design (CAD) models [2]. Indeed, laser scanning technologies allow the collection of large amounts of accurate 3D data. Despite these advantages, space and cost constraints often appear in the use of these types of scanners, which limits their use in building modelling processes, especially when they undergo very frequent updating such as railway stations’ buildings. There is a variety of devices available for the numerisation of 3D data. Scientific literature has often contrived to propose numerous protocols or methods to evaluate the capacity of these new tools, whether in the field of manufacturing, geoscience, geology, botany, or construction. However, it is remarkable to note that with the technological ebullition and new needs, a great number of novel geometric data sensors appear on the market without having at their disposal an adapted analysis protocol to evaluate their performance in precision. Depending on the field of application of the sensor, it is necessary to carry out suitable experimental protocols to evaluate its performance. Today, new technologies such as smartphones or tablets are beginning to be equipped with LiDAR for numerisation purposes. This represents a huge challenge in the future in the numerisation and modelling of buildings. The modelling of interior buildings and the evaluation of the precision and accuracy of 3D data sensors being a vast field, some related works are first presented. The research question of this paper is: can these new LiDAR devices be used in the context of railway stations for lightweight reconstruction? Thus, an experimental approach to evaluate LiDAR on smartphones is proposed in this paper followed by the results and their interpretation; potential improvements not only in scanning capability but also in data acquisition and exploitation are discussed.

A methodological approach to using the smartphone in controlled and field conditions will be proposed to facilitate its handling for a robust numerisation in a building interior environment. The rest of the paper is organised as follows: Section 2 reviews related work. Section 3 presents the material and an approach to evaluate the point clouds of LiDAR smartphones. Results are presented in Section 4. Recommendations for LiDAR smartphone use are discussed in Section 5. Finally, Section 6 is dedicated to the conclusion and future work.

## 2. State of Art

From the collection of geometric data to the representation of a building in the form of a BIM model, a lot of research work has been carried out to establish the format and the model of the use of BIM, as well as to evaluate which tools and analyses are appropriate for reconstructing the condition of a building. Some of this relevant work is presented to introduce our approach to synthesis.

### 2.1. BIM and Scan Technology

BIM in the building life cycle has contributed to several improvements in building design, the construction process, building performance assessment [3,4], operation and management, and indoor environmental monitoring [5]. These different operations are carried out by means of a 3D BIM mock-up [6], which is the digital representation of the building in real time with all the necessary information (geometric and alphanumeric) for its proper management. One of the main difficulties in managing BIM models is keeping them up to date, which is currently done manually and is time-consuming [7,8].

To keep a BIM model up to date, an efficient and flexible method for mapping the interior and exterior environment is needed: The first traditional method is called “CAD to BIM” [9]. In this method, the 3D representation is provided from existing 2D CAD drawings and is designed according to BIM requirements. Software is then used to model the BIM models. This whole method remains quite time-consuming and sometimes difficult to implement due to the absence or lack of up-to-date 2D CAD drawings of existing buildings. The second method is the production method with photogrammetric and measurement techniques using Terrestrial Laser Scanning, LIDAR, and UAV, called ‘Scan-to-BIM’ [9]. Terrestrial laser scanning is the most accurate technique providing millimetre-level accuracy for point clouds captured for BIM models. Point clouds are employed by modellers as a reference from which they manually produce semantically rich BIM models. Another use case of point clouds in the BIM context is to compare them against BIMs to, for example, monitor construction progress or construction (or fabrication) [10,11]. However, terrestrial laser scanners are generally quite expensive and require expertise to use. Also mounted on a tripod, their mapping process in certain areas remains difficult to achieve, for example, overly cluttered rooms and interiors with false ceilings.

With the arrival of new scanners (backpack or hand-held) and some dynamic scanners, there will be a boom in terms of cost and use (within the reach of a wider public). Hand-held scanners equipped on smartphones recently are proving to be an effective new solution for quick and easy point-cloud mapping. With a low cost and the ability to use it in hand, these new scanners on smartphones become a crucial innovation in the development of BIM solutions for more efficient and quicker maintenance of BIM models during their operation, especially on complex buildings such as railway stations. However, to consider the efficient use of these scanners and their data in the context of railway stations, a protocol for the analysis of the sensors and their data is needed. [11]. For point clouds acquired in a BIM context, methodical pre-processing operations are indispensable to guarantee the final quality of the point cloud. However, it is also necessary to consider the various external elements that can affect the point clouds of the scanners’ 3D sensors.

### 2.2. KPI: Indicators for Quality Process of Point Clouds

Defects in a point cloud often come from various sources or factors that affect the laser sensors at the base. Among these, we can cite errors due to the methodology of the survey (e.g., the distance or the angle of incidence [12,13]). In addition to distance and angle, studies have shown that the scanning movement [14,15] and scan time [12] on a surface influence the quality of the scan. To study the accuracy of hand-held scanners on point clouds of industrial workpieces, Gerbinlo et al. [16] and Ameen et al. [17] test the influence of the distance of a hand-held environment and showed how the accuracy of scanners (TLS and hand-held) could be degraded as the range of measurements on flat surfaces increased.

While the factor of distances and the scanning process are mostly known on scanner devices, there are also other external factors influencing scan quality. Literature research has shown that these factors are reflectance, shape, and colour of the object, as well as surface texture and ambient lightning [15,18]. To study the precision of handled scans by the restitution of forms, Lachat et al. [12] and Vogt et al. [14] study and show in front of different forms such as a curvilinear or right shape that we have different precisions of handled LiDAR captors. Moreover, Vogt shows that small geometric forms such as Lego with a small scale do not give a good restitution of object for LiDAR on smartphones.

These properties generate observable defects on a raw point cloud such as: sampling density, noise, outliers, and missing data [3,11]. The distribution of the points sampling the surface is referred to as sampling density. Depending on the use of a scatter plot, a density threshold is necessary. The noise corresponds to the randomly distributed points near the scanned surface. It can be impacted by surface properties, including the scattering characteristics of materials.

The problem of noise and density is due to the distance from the shape to the scanner position, the scanner orientation, as well as the shape’s geometric features [19]. This may be due to the distance from the scanner position, the orientation of the scanner (angle of incidence), as well as the geometric characteristics of the shape [13]. However, most of the studies show that this type of protocol is tested in the manufacturing industry, suitable for testing small objects (artefacts, industrial parts, and indoor equipment). For indoor buildings, context data are missing in the point cloud due to such factors as limited sensor range, high light absorption, and occlusions in the scanning process. A visual analysis makes it possible to identify them qualitatively on the point cloud and outliers are pointed far from the true scanned surface.

### 2.3. Mobile Device Experimentation

In the existing systems of numerisation today, we distinguish several measuring instruments according to the areas of survey and the uses that we want to make of it. There are static systems for geodesy and topography, systems with mobile devices including those rolling on vehicles and those with a handheld device.

To evaluate the accuracy and precision of these sensors, several protocols have been developed depending on the use and the different defects or factors that can affect their data. For a comparative study on the accuracy of a dynamic scanner, Stadia + Osmo, Calantropio et al. [20] offered a comparative study of different static and dynamic scanners while comparing the accuracy of noise measurements and the costs offered by the different materials. Thus, the study focused more on comparing the data with other instruments in an in situ environment. Bolkas and Campus [18] showed how distance, incidence angle, and target-colour can affect point cloud noise on two different sheens (flat and semi-gloss). An experimental protocol was defined to evaluate the factors (distance, angle, colour, time, and repetition) that can generate defects on a point cloud obtained by scanning historic artefacts with a dynamic scanner. In the context of studying other sensors such as the Kinect V2 equipped with a light sensor camera, Lachat et al. [21] highlights the influence of defects such as colour and reflectivity where the Kinect is very often affected by factors such as light and colourimetry on the numerisation of artefact. Sgrenzarol et al. [22] has also shown the use of Mobil scan with Indoor Mobile Mapping Systems (iMMs) for monitoring the progress of buildings during the construction phase. Comparisons have been made between the true depth (Artec Space Spider with Blue Light Technology 1) sensor and the handheld scanner of the iPad Pro. Comparing small Lego shapes, the tests showed that the iPhone LiDAR sensors were not suitable for small objects.

The use of smartphones in point cloud reconstruction is one of the recent new technologies in cloud numerisation. This type of approach is found in early works [23,24] which use smartphone-generated images to texturise points generated by a LiDAR. By simply attaching a protective case with electronics, an integrated Laser Detection Device, and filter, Gao and Peh [25] present the design of a smartphone (Nexus 5)-based laser sensor that allows for an accuracy of 6 cm up to a 5 m distance range on flat illuminated surfaces in outdoor environments. However, the whole system is still non-ergonomic to use, and the acquisition range is still quite low with a small precision. By testing the measurement accuracy on handheld scanners, Luetzenburg et al. [26] made measurements on targets of simple geometric shapes with the LiDAR of the iPhone 12 Pro Max (as part of the numerisation of surfaces in geoscience) to estimate the squared error and the edge effect that can emerge respectively from its sensors in a controlled environment. An analysis of the geolocation, photo-video, and LiDAR sensor systems [27] shows the usefulness of using the iPhone 12 Pro in digitising geological ground surfaces while demonstrating the accuracy shortcomings they can have in geolocation.

In recent works, the LiDAR sensors of the iPad Pro [28] were tested at six different resolutions on building facades in order to evaluate the metrics of the datasets. Comparing these datasets with a static scan in a field environment (open space), the results showed the data from these sensors (processed and high resolution). The same comparison process with terrestrial laser scanners is used to study the point clouds captured by a HoloLens helmet [29] to scan the interior of buildings. The results show a higher coverage of HoloLens compared to the laser scan with however less precision. However, the comparison protocols are only studied in the field environment and do not include studies in the controlled environment. In a recent work, Teppati and al. [30] have evaluated the accuracy and precision of the point cloud of an Apple LiDAR by testing them on different iOS applications (SiteScape, EveryPoint, and 3D Scanner App). Different materials were used depending on the natural sunlight and the results had to show the level of noise and the mode of acquisition with the iOS application. Results show that accuracy could vary depending on the iOS application used. The entire study was carried out in an environment related to buildings in the cultural heritage domain. This work allowed us to highlight the importance of the software component when operating the smartphone LiDAR but a more thorough study in a controlled environment of the LiDAR characteristics (light/surface interaction, geometric primitive, and incidence angle effect) should be highlighted.

### 2.4. Synthesis

In this work, we focus on the analysis of the accuracy, precision, and distribution of the scanning data of the LiDAR smartphone. The analysis of the distribution and accuracy is useful to identify how different types of objects are rendered in 3D by this sensor to anticipate how it can be used in the future in the BIM life cycle management of a railway station building.

A review of published works indicates that the experimental protocols commonly used to test handheld scanners are more suitable for scanning small objects (artefacts, industrial parts, and indoor equipment) (Section 2.3 and Section 2.4). With the advent of new handheld scanners to scan larger areas, some proposed experiments to evaluate sensor data include handheld LiDAR (tablet, smartphone, and Leica BLK2GO). However, the studies carried out on this handled LiDAR do not consider the analysis of several factors that are suitable for testing the behaviour of sensors in an indoor building context, such as the effect of light incidence or the variation of volume as a function of shape and scale. Our experimental approach includes a controlled environment study and a field (in situ) study (where the sensors are tested in different environmental settings). For the in situ tests, we make a comparison of the data captured by the smartphone with those captured with a TLS scanner [28,29,30]. However, in our work, the digitised environments are chosen according to the constraints that the space of the station buildings may present. The aim of the in situ studies will also be to confirm the results obtained in the laboratory while identifying the errors and to show the limitations of using LiDAR in real conditions. The objective is to propose an experimental protocol based on controlled and field environmental [31], testing for the evaluation of new smartphone LiDAR sensors in the context of building interior numerisation. This protocol will allow us to provide recommendations for LiDAR smartphone use for a railway station’s indoor building.

## 3. Material and Method

### 3.1. Material: LiDAR Smartphone: iPhone 12 Pro

LiDAR remote is not a recent invention. In 1962, MIT used it to measure the distance between the Earth and the Moon. Since then, the technology has also proven itself on Apple smartphones and tablets (iPad Pro, iPhone 12 Pro, 13 Pro, and 14 Pro). The integrated LiDAR consists of an array of a vertical-cavity surface-emitting laser (VCSEL) and a direct time-of-flight (dToF) near-infrared (NIR) CMOS image sensor (CIS) with a single photon avalanche diode array (SPAD) [28]. We focused on the iPhone 12 Pro (Figure 1) because of its low cost, its simplicity of use, and its public accessibility (14.6 × 7.1 × 07 cm; height: 186 g; resolution 1284 × 2778 Mpx; panoramic 63 Mpx). Today, to use these LiDAR sensors, there is no native application integrated into Apple devices but rather a multitude of installable iOS applications that allow testing LiDAR sensors. Among these applications, there are two types: those that generate data only in point clouds [32] (e.g., SiteScape), and those that generate data related to the use of meshes derived from the acquired point clouds (e.g., Polycam LiDAR, Roomscan LiDAR, Scaniverse). There are also other applications that are equipped with both functionalities, such as the recently released 3D Scanner App [26]. In the context of our experimentation, we chose to use the 3D Scanner App LiDAR application because of its simplicity of use and the ease with which it generates surfaces with mesh during scanning [33] and the less noisy surface reconstruction [30].

### 3.2. Railway Stations Use Case

The structure of railway stations is made up of not only large spaces, such as train platforms, passenger areas in long corridors, commercial spaces, etc., which are the parts most exposed to the public, but also small spaces, such as offices, technical rooms, rest areas, sanitary spaces, warehouses, or technical building rooms. Thus, the numerisation of structures such as railway stations requires numerous operations using laser scanning tools to generate point clouds. For each situation of maintenance operation which can lead us to numerisation in railway station, we define 5 different use cases (Figure 2) which can lead to geometrical modifications on their BIM mock-up. Use case A (Simple maintenance without a 2D plan): The first case represents work or maintenance modification of small spaces or surfaces which do not need 2D plans. Use case B (Maintenance with the delivery of 2D plans): The second case represents work or maintenance modification of small spaces or surfaces which need 2D plans. Use cases A and B correspond for the most part to maintenance in small spaces. Use case C (Advanced maintenance with the delivery of 2D plans): This case represents work or maintenance modification of larges spaces or surfaces which need 2D plans. Use case D (Maintenance and modification on complex structure): This use case represents the geometric modifications that may occur on structures of a special nature (rail platforms, forecourt, footbridges, etc.). Use cases C and D correspond for the most part to maintenance in large spaces. Use case E (Modification of technical linear network.): The last use case is reserved for experiments in updating linear structures such as piping and HVAC systems which are very complex in station buildings and require a lot of maintenance. In the case of use case E, operations can take place in large spaces (HVAC maintenance in sanitary facilities) as well as in small spaces (repair of electrical conduit on a platform train). Our approach in this paper is to identify if the use of LiDAR smartphones can be a solution to the problems of geometric updating in the context of these use cases.

### 3.3. Experimental Approach to Evaluate the Point Clouds of LiDAR Smartphone of Indoor Building Context: Protocol

Our experimental approach includes a controlled environment study and a field study (in situ). The aim of the in situ studies will be to confirm the results obtained in the laboratory while identifying the errors and show limitations of LiDAR smartphone use in the context of railway stations

#### 3.3.1. Experiments in Controlled Environment

In our first experimental protocol, we will analyse in 3 phases the different factors that can affect the LiDAR smartphone datasets. The first phase (Phase A): we evaluate the behaviour of the data (point clouds) as a function of distance, time, and brightness emitted by a digitised surface based on the work of Lachat et al. [12]. In phase B, to study the effect and the interaction between a light source and the acquisition surface of a digitised object, a design of the experiment was carried out to highlight the interactions between light source and material surface during the numerisation with these new LiDAR sensors. Finally, in phase C, some tests were carried out on different geometries and dimensions to evaluate the behaviour of the digitised data from the LiDAR smartphone. These phases (see Figure 3) allow us to identify the optimal conditions of use of the LiDAR smartphone sensors and to deduce the best use of them in the numerisation of building interiors.


**Phase A: Influence of distance and time**


To study the distance parameter (or acquisition range), a flat material surface is often used on which a contour line is drawn as the measurement sample [12]. For our test, we digitised a 70 × 80 cm contour surface made of wood (diffusely reflective materials) as a sample. The smartphone (LiDAR sensor) was mounted on a tripod and positioned in front of the surface to be scanned at an incidence angle of 0 degrees (Figure 4a). A 6-camera tracking system was used to detect the position of the smartphone relative to the scanned surface (Figure 4b). A system of ball bearings was positioned on the phone to detect it in the tracking space while knowing the position of our surface to be scanned. This device allowed us to carry out the tests more quickly while keeping a good precision on the position of our sensor compared to the digitised surface. The digitised surface is recovered in FORMAT XYZ RGB point cloud data and received under CloudCompare software. The scanned surface samples are fitted with planar primitives in CloudCompare. Plane primitives fitted to the point cloud are a good indicator of the noise level on the point cloud. Once fitted, the normal distance between all sample points in the cloud and the plane primitive can be calculated. The standard deviation of all normal distances characterises the dispersion of the measured points around the plane primitive, which is a good indicator of the overall noise accuracy [12]. The number of points present in each surface sample is also estimated for the calculation of the mean density. For knowing the optimal scanning time [12] to obtain good quality raw data, we studied the time factor in our experiments. For all the tests, the exposure time to scan each surface is 2 s, 5 s, and 8 s. Thus, to estimate the time at which we have stable and usable digitised data, we excluded the numerisation for a time of 1 s since it sometimes takes more than a second for the sensor to start correctly digitising the datasets.


**Phase B: Influence of the material surface and light incidence**


The light intensity is characterised by the point of impact of the light source on the scanned reflective surface. It varies depending on the angle of incidence of the light source with respect to the illuminated surface if this surface is specular or diffuse [18,34]. In order to estimate the impact of the light source and the characteristic surface of the materials on the data provided by the LiDAR sensors, we established a Taguchi experimental design [35,36], to explain the interaction between its two sources of error (factors). To carry out our experimental design, several elements were considered (light incidence, material surface, angle of incidence, distance, and time) for an experimental design on a Taguchi L18 table. The Taguchi L18 table allows for the study of up to 7 factors (including 1 two-level factor and 6 three-level factors) [36]. On the base of our experimentation, we have selected 5 factors for our study; 4 factors have been studied on 3 levels and one factor on two levels.

The choice of the order and levels (Table 1) for our experimental design is as follows: Column 1: The surface material scanned (S) is spread over two levels. The experiment is carried out with two different surfaces: a diffuse surface, Wood S1, and a specular surface, Aluminium S2. Column 2: The light incidence (I) is characterised by the point of impact of the light source on the scanned reflective surface. It varies according to the angle of incidence of the light source in relation to the illuminated surface. In our experiment, we define 3 levels: by an angle of I1 = 0°, I2 = 40°, and I3 = 80° (see Figure 5). Our light source was suspended at a height of 1.30 m from the ground. Column 3: The angle (A) of incidence of the LiDAR acquisition: the smartphone has an angular resolution of 120° (−60°/60° along the LiDAR axis) so we define an angle of incidence between A1 = 0°, A2 = 25°, and A3 = 45°. Column 4: The distance (D) or acquisition range of D1 = 1 m, D2 = 2 m, and D3 = 3 m. Column 5: The acquisition time (T) is set to levels of T1 = 2 s, T2 = 5 s, and T3 = 8 s. For a Taguchi configuration, the factor (source of error) that is supposed to affect the data more should be placed in the first column of the experimental design table [36]. In our experimental design, we want to study the effects and interactions between the incidence of a light source and the scanned material surface while considering other factors that can potentially affect the results during scanning. This explains why we have placed the surface and the light incidence in the first two columns of our table (Table 1) and the other three factors (distances, time, and angle of incidence) in the other columns. In the third column, we placed the angle of incidence because of its major impact on the datasets compared to the distance [13].

For our experimental set-up, we chose two surfaces that reproduce well the two types of light reflection that can be observed in materials: wood—diffuse reflection, and aluminium—specular reflection, on a sample surface of 40 cm × 32.5 cm fixed at 1 m height. A light source (100 W bulb of 4000 K emitting an intensity of 135 Lux) was fixed at a height of 1.30 m and 1 m from our surfaces. During the experiment, we varied the angle of incidence between the surface and the bulb at 0, 40°, and 80°. The objective of the device is to see how the digitised data of a diffuse and specular reflecting surface emerge in front of different angles of light reflection. We used the same tracking system in phase 1 to estimate the position of the lamp, the smartphone in relation to the two scanned wood and aluminium surfaces.


**Phase C: Influence of the Geometry**


To study the parameter of size and geometry, simple geometric primitive shapes such as parallelepipeds, cylinders, and spheres are digitised to study the behaviour in terms of noise and density of the digitised data. These geometries were chosen based on the work of Lachat et al. [12] and Vogt et al. [14], also because these primitives are extremely well represented on station premises equipment, especially in the case of the use case E (Section 3.2). The test samples correspond to geometric shapes of volume types: parallelepipeds, cylinders, and spheres, with dimensions of various scales. The tests will be carried out several times on the same object to assess the accuracy of each shape as we repeat the experiment. Each volume sample is scanned at a fixed distance of 1 m from the sensor. The tests set up considered the repetition factor and the dimensional scale. To evaluate the accuracy of each point cloud, we used the Cloud to primitive distance tool in CloudCompare, which automatically calculates the approximate distance between the cloud and a 3D model with the exact dimensions of each primitive.

#### 3.3.2. In Situ Experiments

In this section, tests are conducted to compare the accuracy level of LiDAR with 3 other reference scanners static and dynamic (see Figure 6.). The most accurate of the three scanners was then selected as a reference for further analysis and comparison with the LiDAR smartphone data in three representative scenes of indoor environments. First was a comparative analysis of the accuracy of the data (Phase 1–2). In a comparison analysis of precision on geometric primitive shapes, two spatial descriptive statistics are considered: sampling noise and deviation (Phase 3). Finally, a histogram of cloud-to-cloud distances is tested to compare smartphone LiDAR data against reference data from the reference scanner (Phase 4). Finally, recommendations are proposed for the use of LiDAR smartphones in the interior of railway station buildings (Phase 5).


**Phase 1: Numerisation**


In this phase, we select a set of scanners with which to make comparisons on our LiDAR smartphone sensor. Based on the work of [19,28,29], we used two TLs scanners and a dynamic scanner. The scanning environment was selected according to the different use cases previously defined in Section 3.2. The room corresponds more to the case of use cases A and B, the warehouse to the case of use case E, and the parking due to its large surface to the case of use case D. Due to the short range of the LiDAR smartphone observed in the laboratory (1–2.5 m for a stable acquisition range), an environment like use case D could not be evaluated.


**Phase 2.1: Target Accuracy Assessment**


The purpose in this phase is to evaluate the accuracy of the LiDAR smartphone compared to other scans in indoor area by using a target measured with a tacheometer [28]. Three other types of scanners were used to compare their results with our new sensor. A Navis VLX (dynamic scan), a FARO X130, and a Riegl scan Vz400i (static scan which generates more noise in an indoor environment). In an indoor scene, 4 targets were placed (see Figure 7) and scanned with our four different scans (see specifications on Table 2)**.** From these 4 targets, the root mean square error is calculated and compared between the 4 scanners to evaluate their level of accuracy compared to measurements that will be made on a tachometer.


**Phase 2.2: Evaluation of geometric primitive’s form: comparison with a controlled environment test**


The second analysis concerns the conformity of the laboratory (seen in Section 3.3.1) and in situ tests in the restitution (Phase 2) of the digital primitives previously digitised during the in situ experiments, confronting real environment conditions with those of a controlled environment. To compare the accuracy of each point cloud, we used the Cloud to primitive distance tool in CloudCompare, which automatically calculates the approximate distance (deviation) between the clouds generated by each sensor and a 3D model with the exact dimensions of each primitive.


**Phase 3: Cloud-to-cloud comparison of FARO vs LiDAR smartphone data**


In this phase, a comparative survey between the LIDAR smartphone and the dynamic scanner FARO Focus is tested. In the context of the railway station use case, we scan 3 types of environments (Figure 8): a parking with 73 m^2^ surface, a warehouse with 34 m^2^, and a local with 13 m^2^. The empty parking was chosen to test the limits of the devices on large surfaces and show the interior luminosity impact on the point cloud data. The storage with its cluttered environment characteristics corresponds to the railway station areas and allows us to evaluate the acquisition capacity in areas overloaded with equipment. The rest of the spaces have been tested according to size, solar lighting, and some equipment similar to the station environment. Based on each environment scanned with the smartphone, we collected different deviations (distributed between 0–2 cm, 2–5 cm, 5–10 cm, and >10 cm) of the points from the FARO static scanner data as reference [29,37,38]. This deviation is determined by using the cloud-to-cloud (C2C) distance analysis on CloudCompare that calculates distance of two-point cloud data with one considered as the reference cloud and the other as the compared cloud. The different deviations that can be observed on the compared cloud (in our case, LiDAR smartphone) come from several factors, namely, the acquisition mode (distance between the sensor and the scanned surface), the geometry of the scanned objects, the overloading of objects, the nature of the material surfaces, and the inaccessibility of some surfaces during the scan.


**Phase 4: Cloud-to-cloud comparison of FARO vs LiDAR smartphone data:**


In this part, a second point cloud comparison analysis is performed considering only some equipment (according to our industrial need) present in the point cloud. We focus on some important parts of the digitised building to assess deviations from FARO data.


**Phase 5: Recommendation**


Based on the results obtained, discussions on the latter will allow us to propose recommendations which are proposed for the use of LiDAR smartphones in the interior of railway station buildings.

## 4. Results

In this section, we present the results of the tests performed in controlled and in situ environments. The conclusions drawn from the first tests allow a better understanding of our sensor for its adaptation to the railway station space.

### 4.1. LiDAR Smartphone Performance Tests in a Controlled Environment

#### 4.1.1. Influence of Distance and Time on Data (Phase A)

The standard deviation measurements on the distances (Section 3.3.1) allow us to evaluate up to which distance we can obtain a stable and non-critical noise level to determine in which distance interval we can obtain optimal data. The overall results (see Figure 9a) show a slight increase in standard deviation between distances of 1 m and 2.5 m, but from 2.5 m onwards, a significant increase can be seen. The measurements were carried out for the series of tests over distances of 1 to 3 m. The density (between 28,767 pts/m^2^ and 19,148 pts/m^2^ Figure 9b) decreases as the range increases but remains relatively stable without significant variation at longer ranges. Nevertheless, the density remains high for a small range and a long acquisition time.

#### 4.1.2. Influence of the Material Surface and Light Incidence (Phase B)

The Table 3 lists the different test configurations (Figure 5.) to be performed with the results where Y represents the values of the standard deviation of the noise of the scanned surfaces for each test configuration of the experimental design. The lowest response of Y corresponds to test L11 with a standard deviation of 1.15 mm while the highest response of Y corresponds to test L14 with a value of 6.8 mm as standard deviation. From this configuration, we calculated the effect of the factors and the interaction between the surface and the light intensity.

Formula of the Effects and Interactions:
Effect of a factor *S* at level *i = ESi = MeanYSi − M*
(1)

where *MeanYSi* is the mean of the *Y* values for *S* at *Si* and *M* the mean of *Y* values.

Effect of a factor *I* at level *i = EIj = MeanYIi − M*
(2)

where *MeanYIj* is the mean of the *Y* values for *I* at *Ij* and *M* the mean of *Y* values.

Interaction between two factors *S* at level *i* and *I* at level *j = ISiIj = MeanYSiIj − M − ESi − EIj*
(3)

where *MeanYSiIj* is the mean of the *Y* values for *S* at the level *i* and *I* at the level *j*.

On the graphs of the effects of each factor (surface and light intensity) (Figure 10a), we observe that the factor with the most influence on the data is the distance and the one with the least influence is the surface of materials. However, we can see a strong interaction between the “surface (factor *S*)” and the “light incidence (factor *I*)” when calculating the interaction effects between the two factors (as the lines cross each other strongly on the interaction graph; see Figure 10b). Especially when the surface is specular (S at level 2; Aluminium) and the light is at an incidence of 40° (Response Y in Table 3), the interaction is even stronger. A visual analysis for this test case shows an intense reflection of light from the surface at the time of scanning (see Figure 11b). The best configuration of the model to have minimum noise is to set the configuration where all effects and the interaction between S and I are at a minimum. We can deduce that the best configuration to obtain low noise is the one where we scan a wooden surface (S1) with a light incidence in I2, a distance or range of 1 m (D1), an incident angle of 0° (A1), and an optimal scanning time of 8 s (T3). The effect of distance (Table 4) is quite large on the behaviour of the LiDAR data and that of surfaces is small. However, in the presence of a diffuse surface (level 1), there is a very strong interaction with the light incidence which can disturb the output measurement noise.

#### 4.1.3. Influence of the Geometry (Phase C)

Acquisition of a parallelepiped shape: The test is based on the work of [12,14,24] where the dimensions of the sections are compared, knowing that the sections are made in point clouds from repeated acquisitions. For this test, we scanned in point clouds two parallelepiped boxes P1 and P2 made of cardboard with external dimensions of about 55× 55 × 33.6 cm for P1 and 46 × 36 × 27 cm for P2 (Figure 12a,b). They were scanned successively under the same conditions, moving the scanner around the object at a nearly fixed distance of 1 m. To analyse the accuracy of the data, primitives corresponding to the real dimensions were fitted to the point clouds of the two scanned parallelepipeds P1 and P2. A root mean square error (for a series of five tests performed) was evaluated over 8 mm and 13.44 mm concerning P1 and P2 point clouds. The noise dispersion calculated from the primitive fitted (Cloud to primitive) gives a standard deviation of about 7.2 mm for P1 and 7.4 mm for P2, with a general mean distance point to primitive at 8 mm and 13.1 mm, respectively, for P1 and P2.

Effect of measurement acquisition repetition: For further analysis on the aspect of measurement repeatability, we scanned the P1 card by performing a one-turn, two-turn, three-turn, and then four-turn displacement relative to the object. At each turn, we performed three trials. This will allow us to evaluate how the dataset behaves when scanning the same surface or volume repeatedly. A root mean square error (for a series of three tests performed) has been calculated and results in values of 8–8.12–8.44 and 10.74 mm for the one-turn-two-turn-three-turn-four-turn scans. The results show that the LiDAR smartphone provides data that degrade as the scanning is repeated over the same area or volume (especially beyond two repetitions).

Acquisition of a cylindrical shape: For this study, two cylindrically shaped PVC objects were scanned by moving our LiDAR smartphone around at an almost fixed distance of 1 m. The first cylinder (C1) has a bigger diameter of 40 cm for a height of 50 cm and the second (C2) (Figure 12c,d) has a smallest diameter of 20 cm for a height of 130 cm. Using the open-source point cloud processing software CloudCompare as in phases 1 and 2, we created cylindrical primitives to fit into the point clouds of the scanned cylinders. For cylinders C1 and C2, five tests were performed; the calculated radius of the fitted primitive varies between 20.2 and 20.5 cm for C1 and between 10.6 and 10.9 cm for C2, which represents a root mean square error of 4.42 mm and 8.63 mm compared to the real dimensions of the object. Error precision increases much more in the presence of a low-dimensional cylinder radius and the aberrations (see Figure 12c,d) are not negligible for these types of geometry. Moreover, a visual analysis allows us to evaluate an important edge effect on the end of the cylinders. The noise dispersion calculated from the primitive gives a standard deviation of about 5.8 mm for C1 and 10 mm for C2, with a general mean distance point to primitive fitted (Cloud to primitive) at 25 mm and 25.3 mm, respectively, for C1 and C2.

Acquisition of a spherical shape: For this study, a plastic ball with a spherical shape (Sp) was scanned by moving our LiDAR smartphone at 1 m. The sphere is 55 cm (27.5 cm) in diameter (see Figure 12e). Using the free point cloud processing software CloudCompare as in phases A and B, a spherical primitive was fitted into the sphere point clouds. For the sphere, five tests were performed; the calculated radius of the fitted primitive varies between 28.1 cm and 29 cm, which represents a root mean square error of the deviations of 1.4 mm from the real dimensions of the object (27.5 cm). The noise dispersion calculated from the primitive gives a standard deviation of about 6 mm, with a mean distance point to primitive of 13.7 mm.

### 4.2. In Situ Experiments

#### 4.2.1. Comparison of LiDAR Smartphone with Navis, RIEGEL, and FARO Scanner: Target Accuracy Assessment (Phase 2.1)

From the workflow described (in Section 3.3.2) with a target (t_1_, t_2_, t_3,_ and t_4_), the successive distances (d_12_; d_23_; d_34_; and d_41_) between the four targets are calculated with a Leica tacheometer measurement. Those distance are also measured with the Leica, and the point cloud measurements of each sensor used. Once the distances were known, a root mean square error was calculated for the distances of each sensor in the Table 5. A variation of the RMSEs for the selected sensors is observed. The distances closest to that of the tacheometer are those of the FARO. For the smartphone LiDAR, there is a non-homogeneous distribution of the MSE as the scanned area gets larger. It can be deduced that the accuracy of the smartphone LiDAR data degrades very quickly for large areas.

#### 4.2.2. Comparison of LiDAR Smartphone with Navis, RIEGEL, and FARO Scanner: Evaluation of Geometric Primitive’s Form (Phase 2.2)

Once the deviations of each of the points are known, we can estimate the mean of all these deviations and their standard deviation. The results show that the FARO data has less deviation compared to the Navis and LiDAR sensors. The VZ 400i data has less accurate results because it is a static time-of-flight sensor, and it tends to have a lot of noise in an indoor space. The point clouds of all three environments were collected not only from the smartphone LiDAR scanner, but also from the FARO Focus X130. For the LiDAR smartphone sensor, the standard deviations are very large as we go along for small-scale geometries (c1 cylinder and P2 parallelepiped) (Figure 13a). This confirms the results observed in the laboratory (Section 4.1.3). However, we also observe a larger deviation in situ because all objects were scanned together and not individually as in the laboratory (Figure 14a). On the other hand, the distance between the point clouds and the primitives is much more important for the smartphone sensor when curvilinear shapes are present (8 mm sphere and 15–22 mm cylinder). This mean distance is less important with parallelepipeds (6–7 mm): Figure 14b.

From the test observed in the first part, the results allowed us to show that the FARO Focus X130 sensors are the most accurate and present the results closest to the exact data. This will allow a comparison of the two datasets to estimate the degree of accuracy of the smartphone LiDAR compared to the FARO.

#### 4.2.3. Comparison of Environment Quality of Reconstruction (LiDAR Smartphone vs. FARO): (Phase 3)

In this workflow, specific zones of the LiDAR smartphone point cloud with a significant deviation from the FARO reference cloud will be referred to as the critical zone. This zone will correspond to points located on a deviation of more than 5 cm (industrial specification). Overlapping of surfaces or two passes over the same surface after a long waiting time generates overlapping layer effects. The device and the acquisition environment allowed us to perform a single acquisition for the LiDAR smartphone for the FARO dynamic scanner. The acquisition was carried out using two to three stations depending on the space and the difficulty of accessing certain surfaces. The results of these experiments are presented in the Figure 15, where the colours represent the deviation scales obtained between the smartphone LiDAR cloud and the FARO cloud considered as the reference cloud.

For the Parking, the data loses accuracy as a function of the area (deviations are greater in the second half of the scanned area); the main critical areas of the smartphone point clouds were located along the wall, the geometric targets, the wire fence area, and some small equipment. Globally, 78% of the points has <5 cm of deviation. Surfaces at the top are less dense than those at the bottom. Part of the parking covered by a grid surface generates significant reconstruction errors compared to the FARO data (deviation of 15 cm–27 cm) Figure 15a.

For the warehouse, the main critical areas (Figure 15b) of the LiDAR smartphone point clouds are located at the level of the shelves where there is an overload of technical equipment. Due to a high presence of clutter in the warehouse, a significant deviation is observed in the digitised environment. (21% of the points, >5 cm of deviation). Analysis of openings such as the front door shows a deviation of 0–2 cm but which turns into 2–5 cm on the edges. The LiDAR of the smartphone generates much more noise on surface edges. However, in this environment, the LiDAR smartphone can scan more inaccessible areas than the FARO static scanner. Equipment such as walls, floors, openings, and fixed furniture are under 5 cm of deviation. The important noise and deviation observed in the data show that clutter increases the inaccuracy of the data (50% of the surface is cluttered).

For local, the main critical zone of the smartphone point clouds is located at the corners of objects, hard-to-survey wall areas, and small-scale geometries. Analysis shows that regions with a light lamp have a significant deviation (which confirmed the influence of surfaces and light observed in Section 4.1.2: 89% of the points, <5 cm of deviation). Some aberrations (Figure 15c) are located on glass surfaces, but the small dimension of the space (13 m^2^) allowed us to have a better precision of the data of the smartphone compared to the FARO.

#### 4.2.4. Parametric Comparison of FARO vs. LiDAR Smartphone Data (Phase 4)

The analysis focused on the surface elements shows an increase in the percentages of points below 5 cm of deviation for the local and the warehouse. However, for the parking data, the number of points below 5 cm of deviation did not change due to the small space in the local and the presence of the wire fence which generates a strong deviation. If only the surface elements are considered (wall, ceiling, floor, and beam ledge), the LiDAR smartphone data increases in accuracy. The analysis focused on openings (doors and windows) shows that the percentage of points below 5 cm deviation are mostly above 80%. However, openings with glass surfaces generate significant artefacts near the edges of the opening. Edge effects are observed at the edges of openings. Fixed equipment in the scanned environments was also analysed on large vending machines, waste bins, and a cylindrical water cooler. The deviations obtained are of the order of 98% of the points above 5 cm accuracy for the dispensers; 96% of points below 5 cm accuracy for waste bins; and 72% of points below 5 cm for the cylindrical water cooler.

## 5. Discussions: Recommendation for the Use of LiDAR Technologies in Railway Context

**Surface acquisition:** Overall, the plane-fitting investigation showed a local accuracy of 1 to 3 mm and an average density around 19,000–20,000 pts/m^2^ for the LiDAR smartphone data use with the 3D Scanner application. These accuracy levels are also observed in studies conducted with the same type of LiDAR on the iPad Pro in open indoor spaces by using the same application [27,28,30]. However, these are still relative accuracies that are highly dependent on the digitised environment. Indeed, our approach has shown that these accuracy values can quickly degrade depending on the dimensions of the digitised space (parking) and the encumbrance (warehouse). The results obtained in the target study show that the LiDAR sensor of our smartphone loses accuracy as the space increases. The deviations measured in Section 4.2.2 show that our tested sensor is increasingly more accurate for the local environment than the warehouse and the parking. The tests in the parking also showed that, from the second half of the scanned area, the deviations (FARO-LiDAR smartphone) increased from 0–2 cm to 2–4 cm. As the acquisition areas increase, the accuracy errors increase. This is confirmed by comparing the percentage deviation (FARO-LiDAR smartphone) <5 cm between the largest and smallest environment (rep 78% for the parking, 79% for the warehouse, and 89% for the local) whereas existing work on iPad and iPhone 12 LiDAR sensors shows percentage deviation measurements (FARO-LiDAR smartphone) of more than 85% on outdoor and indoor scans [28,30]. The loss of accuracy with increasing space is also observed in the measurement tests in 4.2.1 where the accuracy deteriorated as the area scanned increased.

The LiDAR smartphone would be recommended for areas up to 40 m^2^ with a footprint of less than 50% of the area to be scanned (ideal for use cases A and B). This limits the use of the sensor in the case of use cases C and D (see Section 3.2).

**Optimal acquisition range:** With the results observed in a controlled environment (Section 4.1.1), the maximum order of magnitude for the acquisition range is estimated to be 2 m. This may seem very small compared to traditional static scanners [12,13], but the low weight and design make it easy to handle for scanning hard-to-scan areas.

During the operation of acquisition, the tests allowed us to observe that the data were less disturbed (deviation in parking) if we avoid scanning in certain parts of the equipment and ensure the movement with iPhone 12 around the local being scanned without returning to re-scan the areas already scanned because that can cause the problem of double surface like in case of the parking and the storage.

**Risk areas:** The result in phase B and the cloud-to-cloud deviation measurements (LiDAR vs. FARO smartphone) allowed the identification of risk areas such as overly reflective surfaces (Taguchi experiment) and gridded surfaces which generate a large accuracy error with the LiDAR smartphone (see Phase 3 Figure 15a). An interesting result of this study is that reflected diffuse surfaces affect noise much less than specular surfaces (e.g., 3.6 mm standard deviation vs. 6.8 mm). The surface rendering analysis showed that the LiDAR data lost three times the accuracy for specular reflective surfaces and semi-transparent surfaces. Smartphone LiDAR degrades the data much more compared to traditional LiDAR which degrades the surface little under the same conditions [18]. This can lead to digitised environments with large deviations in areas that are in the same state as these surfaces. This loss of performance can be explained by the type of sensor used in the phone and its poor ability to scan surfaces in the same way as the usual LiDAR.

**Geometric form available:** It is also worth noting that the tests for the restitution of primitive geometric shapes gave interesting results for the recovery of global dimensions of geometric shapes. Compared to sensors such as the FARO freestyle or true depth scanner [12,14], the iPhone 12 Pro LiDAR offers a lower accuracy [12]. However, these values allow their potential use for the scanning of certain geometric shapes that do not have too-small dimensions [14].

However, some problems common to handheld scanners, such as edge effects and high deflection, may limit their use when high-precision acquisitions are required. Moreover, during the tests on the geometric shapes, the results showed that the dimensional scales had an impact on the accuracy of the geometries: a standard deviation, mean of 5.8 mm, 22.2 mm for a large diameter of C1 versus 14 mm, 15 mm for the small diameter of cylinder C2. We have the same observation with the large parallelepiped P1 (6.6 mm, 6.5 mm) against 19.4 mm, 7.8 mm for the small parallelepiped P2.

In conclusion, with the in situ and laboratory results, the loss of accuracy of the LiDAR smartphone data for small-scale geometric primitives does not allow the use of the sensor to be recommended in environments with a significant presence of linear engineering networks (see use case E). Equipment such as walls, doors, openings, ceilings, floors, fixed furniture, and other equipment larger than 20 cm in diameter or larger than a volume of 0.5 × 0.5 × 0.5 cm^3^ are recommended for scanning due to their low error rate (80% below 5 cm of deviations).

## 6. Conclusions and Future Works

In this paper, we investigated the spatial mapping capability of the iPhone 12 Pro LiDAR in the environment of railway station interiors. We performed a geometric evaluation of the indoor data on acquisition distance, reflection, and time, while subsequently moving on to a comparison of the sensor with other scanners in a test environment like that of a railway station. Comparison of the smartphone LiDAR data with a point cloud from the laser scanner and a reference 3D model of the test environment showed that the LiDAR smartphone mesh is broadly correct, over environments equivalent to the use cases A and B defined in Section 2.4. Overall, the experimental results indicate the significant benefit of using these sensors in building mapping. Based on the results obtained, we were also able to propose recommendations for the intervention of LiDAR for numerisation in certain categories of railway station spaces (use cases A and B).

Through this paper, our experiments have shown that scanning from the LiDAR sensor of the smartphone presents some constraints like those of other handheld scanners such as an increasing increase in the noise of the data measurement as its range and angle of incidence increases. As for the time required to scan a surface, our tests showed that this parameter had little effect on the scanned data. The analysis of geometric shapes from the smartphone LiDAR in paragraph 5 showed us that primitives with finite edges (parallelepipeds and cylinders) generated point clouds with significant measurement noise, especially at the edges, and accuracy errors (1 to 2.5 cm). By varying the dimensions of the parallelepiped and cylinder primitives, we found that the noise and accuracy increased as the size of the scanned volume primitives decreased in dimensional scale (Section 3.3.1 Parallelepiped P1: 55× 55 × 33.6 cm and P2: 46 × 36 × 27 cm, Cylinder C1 with 40 cm diameter and C2 with 20 cm diameter).

To consider the potential use of the LiDAR smartphone to the scan indoor buildings, a certain amount of care must be taken during the scanning process, such as avoiding jerky, accelerated, or abrupt movements in order not to lose information and accuracy. The tests also made it possible to show the usefulness of the smartphone in the context of use case A and B while showing an accuracy of the order of 4 to 5 cm. Recommendations for using the smartphone LiDAR presented in Section 5 will allow us to focus on the main equipment that can be scanned in our future 3D model update process. In future works, we will investigate how to root the transition from point clouds to CAD models [3,39,40,41,42]. It will be interesting to test which of the major segmentation methods are best suited for the data collected by these types of point clouds.

With the new LiDAR smartphones such as the iPhone 13 Pro [43] and 14 Pro with more resolution, we will be able to adapt our protocol on these new sensors due to their similarity with the iPhone 12 Pro. The aim of this test will be to evaluate the improvements that the new sensors can bring to the digitisation of railway station areas. We are also looking at possible improvements to the scanning methods, and we are considering testing new iOS scanning applications that will be available in the future.

## Figures and Tables

**Figure 1 sensors-23-01967-f001:**
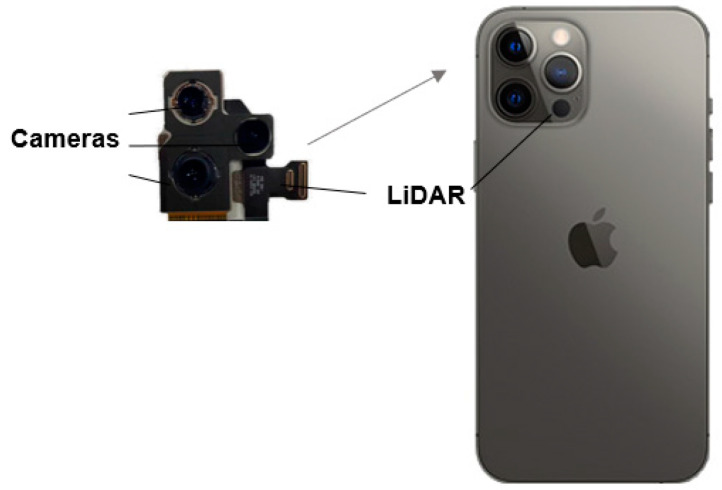
LiDAR system on an iPhone 12 Pro Max.

**Figure 2 sensors-23-01967-f002:**
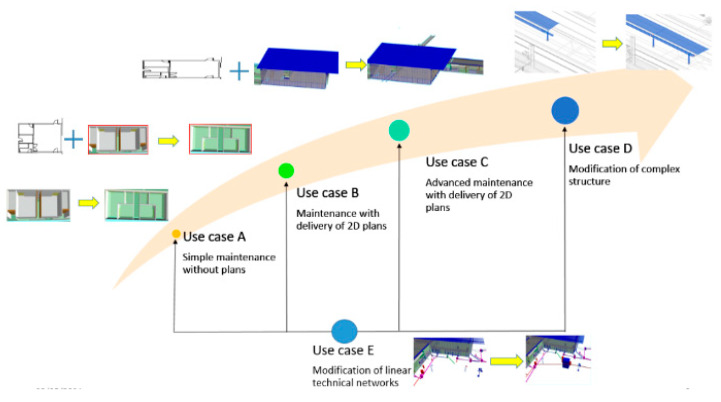
Different use cases of the experimentation presented in several situations of maintenance operations.

**Figure 3 sensors-23-01967-f003:**
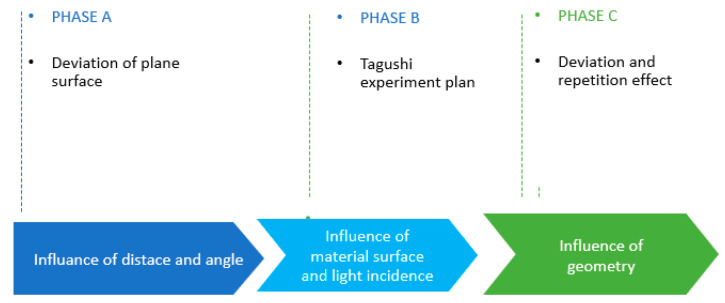
Laboratory Experimental Measurement Protocol.

**Figure 4 sensors-23-01967-f004:**
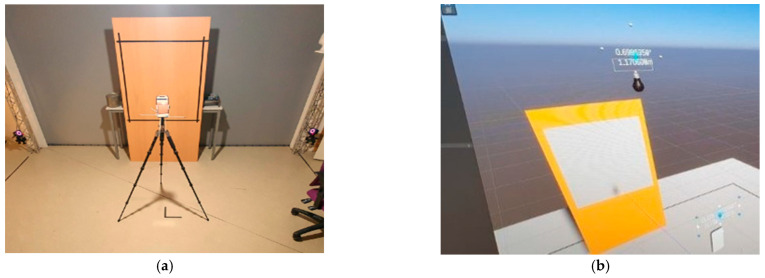
Distance–time measuring device: (**a**) Assembly of the device with the LiDAR smartphone iPhone 12 Pro Max and the scanning surface; (**b**) Interface of the tracking device to estimate the position of the sensor relative to the surface:.

**Figure 5 sensors-23-01967-f005:**
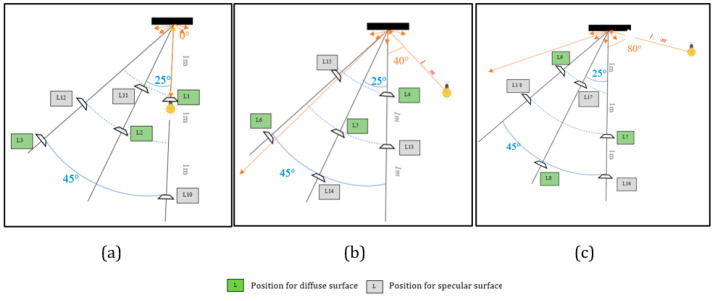
Different schematic configurations in view of the selected Taguchi plane L ()18: L represents the LiDAR position for each trial. (**a**) Different position of the sensor depending on the light incidence at 0°. (**b**) Different position of the sensor according to the light incidence at 40°. (**c**) Different position of the sensor in function of the light incidence at 80°.

**Figure 6 sensors-23-01967-f006:**
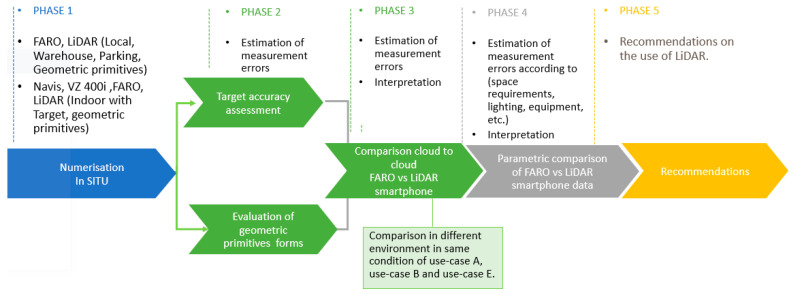
In Situ Experimental Protocol.

**Figure 7 sensors-23-01967-f007:**
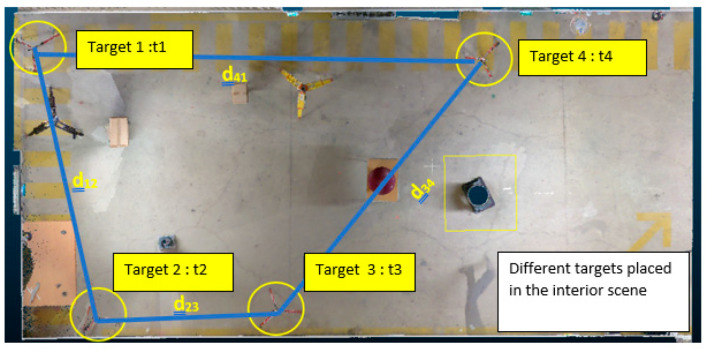
Different targets in the indoor environment.

**Figure 8 sensors-23-01967-f008:**
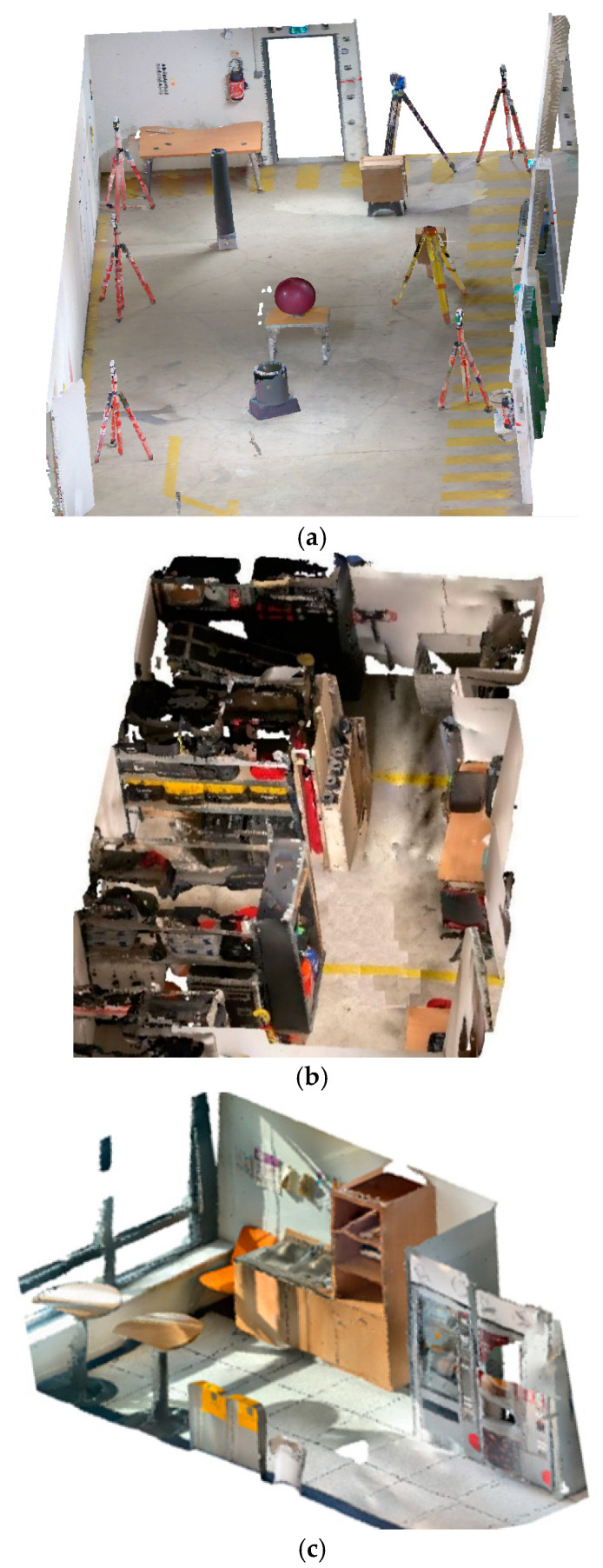
Perspective image of the meshed point cloud for (**a**) the parking, (**b**) the warehouse, and (**c**) the local.

**Figure 9 sensors-23-01967-f009:**
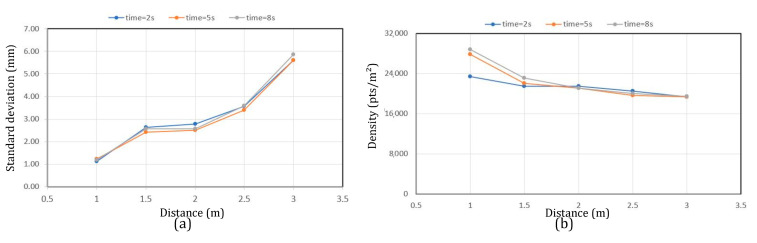
(**a**) Standard deviation of noise as a function of time and distance. (**b**) Density of points (points/m^2^) as a function of time and distance.

**Figure 10 sensors-23-01967-f010:**
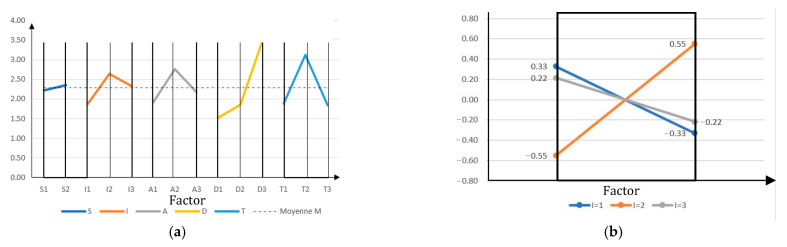
(**a**) Graph of factor effects. (**b**) Graph of intensity between factors S and I.

**Figure 11 sensors-23-01967-f011:**
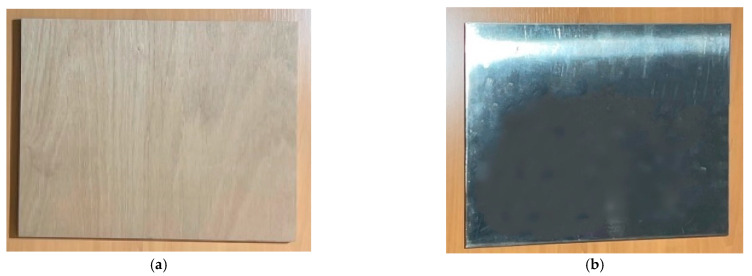
(**a**) Wooden plate (diffuse reflection) and (**b**) aluminium plate (specular reflection): the visual impact shows the strong light reflection of a specular surface compared to a diffuse surface.

**Figure 12 sensors-23-01967-f012:**
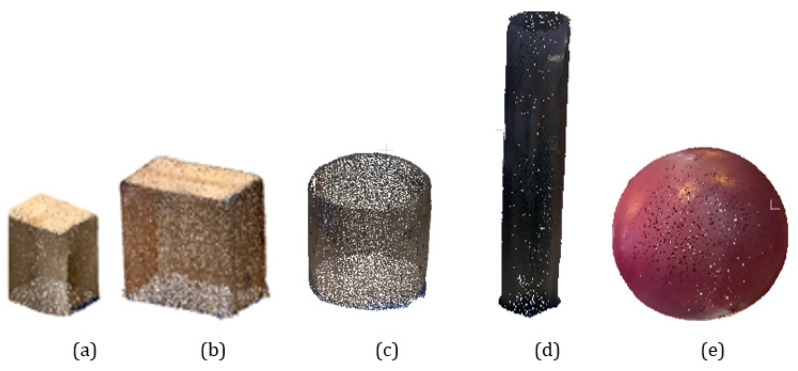
Point cloud parallelepiped shape: (**a**) *P1;* (**b**) *P2* of the experimental cylinders. Point cloud of the experimental cylinders *C1* in (**c**) and *C2* in (**d**); Point clouds of digitised sphere *Sp* in (**e**).

**Figure 13 sensors-23-01967-f013:**
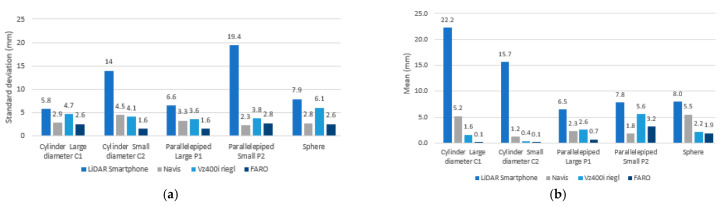
(**a**) Cloud–primitive distance standard deviation diagram of the 4 scanners in situ on geometric primitives. (**b**) Diagram of the mean distance Cloud–primitive of the 4 scanners in situ on geometric primitives on geometric primitives.

**Figure 14 sensors-23-01967-f014:**
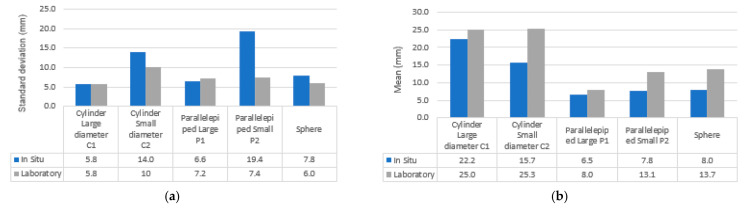
Comparison in situ vs Labo of LiDAR smartphone data on geometric primitives: (**a**) for standard deviation Cloud to primitive; (**b**) for the mean distance Cloud to primitive.

**Figure 15 sensors-23-01967-f015:**
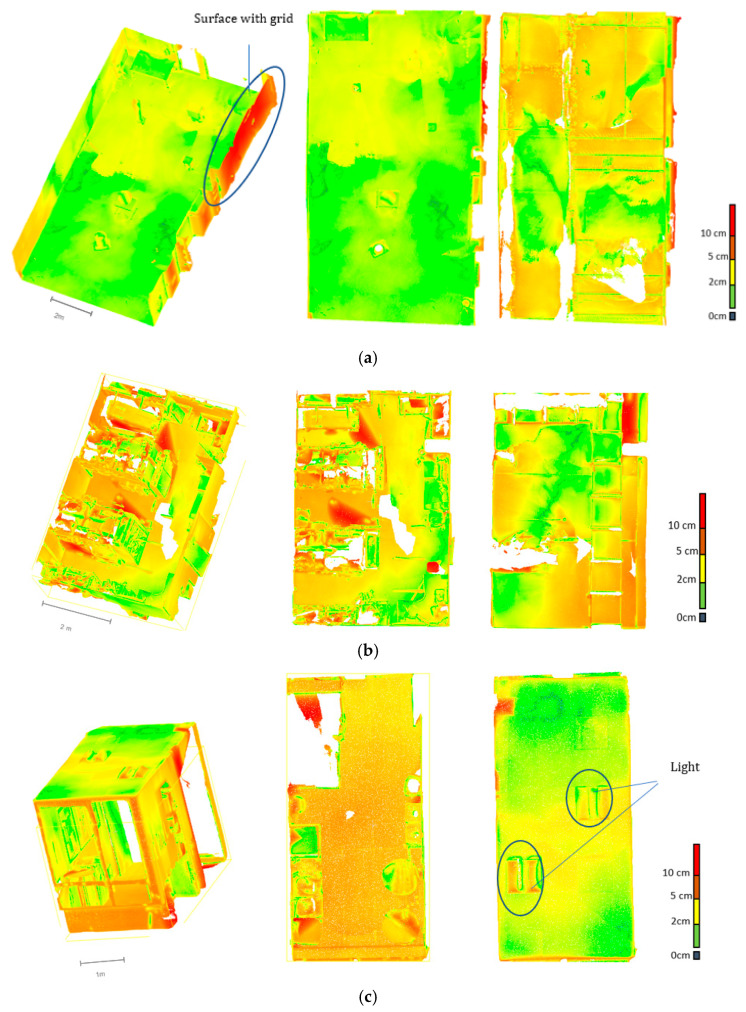
Profile view, ground plane, and ceiling plane showing the deviation map of the LiDAR point cloud from the FARO reference cloud: (**a**) Parking; (**b**) Warehouse; (**c**) local. Colour green is with a deviation of 0–2 cm, yellow with a deviation of 2–5 cm, orange with a deviation of 5–10 cm, and red with a deviation greater than 10 cm. Environments with areas such as cluttered technical rooms, rooms with glass or semi-transparent materials, rooms with intense lighting and reflective surfaces, and outdoor open space should be avoided.

**Table 1 sensors-23-01967-t001:** Taguchi table L18.

	Column 1	Column 2	Column 3	Column 4	Column 5
Tests	Surface	Incidence Of light	Angle	Distance	Time
L1	S1 = Wood	I1 = 0°	A1 = 0°	D1 = 1 m	T1 = 2 s
L2	S1 = Wood	I1 = 0°	A2 = 25°	D2 = 2 m	T2 = 5 s
L3	S1 = Wood	I1 = 0°	A3 = 45°	D3 = 3 m	T3 = 8 s
L4	S1 = Wood	I2 = 40°	A1 = 0°	D1 = 1 m	T2 = 5 s
L5	S1 = Wood	I2 = 40°	A2 = 25°	D2 = 2 m	T3 = 8 s
L6	S1 = Wood	I2 = 40°	A3 = 45°	D3 = 3 m	T1 = 2 s
L7	S1 = Wood	I3 = 80°	A1 = 0°	D2 = 2 m	T1 = 2 s
L8	S1 = Wood	I3 = 80°	A2 = 25°	D3 = 3 m	T2 = 5 s
L9	S1 = Wood	I3 = 80°	A3 = 45°	D1 = 1 m	T3 = 8 s
L10	S2 = Aluminium	I1 = 0°	A1 = 0°	D3 = 3 m	T3 = 8 s
L11	S2 = Aluminium	I1 = 0°	A2 = 25°	D1 = 1 m	T1 = 2 s
L12	S2 = Aluminium	I1 = 0°	A3 = 45°	D2 = 2 m	T2 = 5 s
L13	S2 = Aluminium	I2 = 40°	A1 = 0°	D2 = 2 m	T1 = 2 s
L14	S2 = Aluminium	I2 = 40°	A2 = 25°	D3 = 3 m	T2 = 5 s
L15	S2 = Aluminium	I2 = 40°	A3 = 45°	D1 = 1 m	T3 = 8 s
L16	S2 = Aluminium	I3 = 80°	A1 = 0°	D3 = 3 m	T2 = 5 s
L17	S2 = Aluminium	I3 = 80°	A2 = 25°	D1 = 1 m	T3 = 8 s
L18	S2 = Aluminium	I3 = 80°	A3 = 45°	D2 = 2 m	T1 = 2 s

**Table 2 sensors-23-01967-t002:** Specifications for the LiDAR smartphone, the Navis VLX, the Vz 400i, and the FARO Focus X130.

	LiDAR Smartphone iPhone 12 Pro Max	Navis VLX	Vz 400i RIEGL	FARO Focus X130
**Range**	0.5–5 m	0.5–60 m	0.5–800 m	0.6–130 m
**Precision**	5–20 mm	6 mm	1 mm	2 mm
**Size**	14.6 cm × 7.1 cm × 0.7 cm	108 cm × 33 cm × 56 cm	30 cm × 22 cm × 26 cm	24 cm × 20 cm × 10 cm
**Type of scan**	Dynamic	Dynamic	Static	Static
**Sensor**	TOF: Time Of Flight	TOF: Time Of Flight	TOF: Time Of Flight	TOF: Time Of Flight

**Table 3 sensors-23-01967-t003:** Standard deviation response Y of the plan surface scanned.

Test	Y (mm)
L1	1.53
L2	1.78
L3	3.10
L4	1.81
L5	1.55
L6	2.68
L7	2.02
L8	3.69
L9	1.75
L10	1.76
L11	1.15
L12	1.91
L13	1.64
L14	6.80
L15	1.31
L16	2.70
L17	1.60
L18	2.27

**Table 4 sensors-23-01967-t004:** Effect of different factors in the tests (mm).

Factors	S	I	A	D	T
**Effect of level 1**	−0.07	−0.41	−0.37	−0.76	−0.40
**Effect of level 2**	0.07	0.35	0.48	−0.42	0.84
**Effect of level 3**		0.06	−0.11	1.18	−0.44

**Table 5 sensors-23-01967-t005:** Root mean square error of distance data d12, d23, d34d, and d41 from VZ, Navis, and FARO sensors.

	MSE: Mean Square Error (m)			
	**d_12_**	**d_23_**	**d_34_**	**d_41_**
**LiDAR Smartphone**	0.037	0.321	0.100	0.090
**Navis**	0.005	0.005	0.001	0.007
**VZ**	0.007	0.003	0.001	0.003
**FARO**	0.001	0.007	0.004	0.004

## Data Availability

Data available upon request from the authors.

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
