# Peer review of "Smartphone LiDAR Data: A Case Study for Numerisation of Indoor Buildings in Railway Stations"

_sensors, 2023, doi:10.3390/s23041967_

Round 1

Reviewer 1 Report

This manuscript is heavily short of innovation. It appears that this manuscipt is  technical report. 

Author Response

Manuscript ID

 sensors-1958684

Title:

 Smartphone LiDAR Data: A Case Study for Numerisation Of Indoor Buildings In Railways’ Stations.

The combination of LiDAR with other technologies for numerisation is increasingly applied in the field of building, design, or geoscience, as it often brings time and cost advantages in 3D data survey processes. In this paper the reconstruction of 3D point cloud datasets is studied by an experimental protocol evaluation of new LiDAR sensors on smartphones. To evaluate and analyse the 3D point cloud datasets, different experimental trials are considered depending on the acquisition mode and the type of object or surface being scanned. The conditions allowing to obtain the most accurate data are identified to propose which acquisition protocol that seems to be the most adapted when using these LiDAR sensors to digitise complex interiors building such as railway stations. The aim of this paper is to propose: (i) a methodology to suggest the adaptation of an experiment protocol based on factors (distance, luminosity, surface, time , incidence) to assess the precision and accuracy of the smartphone LiDAR sensor in a controlled environment; (ii) a comparison, both qualitative and quantitative, of Smartphone LiDAR data with other traditional 3D scanner alternatives (Faro X130, VLX, Vz400i) while considering 3 representative building interior environments; and (iii) a discussion of the results obtained in a controlled and a field environment , making it possible to propose recommendations for the use of the LiDAR smartphone at the end of the numerisation  of the interior space of a building.

Comments for the reviewers:

First, we would like to thank you for your relevant comments and suggestions.

Please find a table listing our responses (in red) to comments (in black). Each response is structured in two steps:

- A detailed response to the reviewer’s comment in a small paragraph.

- A short summary of changes made to the paper with a reference to the page and line number.

Please find attached an updated version of our paper. Alterations resulting from the reviewer 1 (resp. reviewer 2, resp. reviewer 3) comments are highlighted in yellow (resp. green, resp. blue).

We hope this new version will meet your expectations.

Yours faithfully,

The authors.

Response to reviewer 1:

Point 1: This manuscript is heavily short of innovation. It appears that this manuscript is technical report.

Overall response:

The authors thank the reviewers for their comments and interest in our work.

We hope that the new arguments we will bring to support and sustain the scientific and academic relevance of our work will convince you to accept our paper.

Response 1:

The work presented in our paper in our paper aims to evaluate the use of lidar in a controlled test case and in a real case and compares it with TLS equipment to assess its suitability for use in railway station environments. The final purpose of the work is to identify a new device of 3D scan that will facilitate the input of geometric data for the updating of BIM models of railway stations.

These studies were based on numerous parallel scientific works carried out to test the accuracy of 3D environmental sensors in an industrial context (Buildings; Automobiles; Manufacturing industry).

(Page 2, line4-13 line15-23)

The experiments presented to test the performance of these LiDAR’s on smartphones were carried out based on tests already available in the scientific literature.

In fact, with reference to the tests carried out for the characterisation of LiDAR sensors, the experimental tests carried out in the laboratory had the following objectives:

- to characterise the level of accuracy of the smartphone LiDAR as a function of the acquisition range, the angle of incidence and the primitive geometric shapes (cylinder, parallelepiped, etc.) with reference to the tests carried out for the characterisation of LiDAR sensors

- study the influence of the scanning time on the surface of an object (flat surface made of wood and aluminium),

- to highlight the influence of the interaction between the luminosity and the scanned material surface (specular surface in aluminium, specular surface: wood).

(Page 3, line31-39,line41-48)

Building on previous work on LiDAR and setting our industrial context, we have proposed a comprehensive approach that shows the importance of laboratory testing and how it can facilitate the performance and understanding of in situ testing. This has not been demonstrated for the use of smartphone LiDAR for building digitisation.

(Page 5, line 40-53)

The In-SITU experiment was carried out in environments like those of a railway station interior. They aimed to compare the data digitised by the LiDAR smartphone with 3 other traditional scanners (Faro X130, Navis VLX, Reigl Vz 400i) to evaluate their accuracy in relation to the latter. The most accurate scanner was selected for direct comparison with the LiDAR data. The originality of the work comes from the experimental approach implemented according to the different parameters in a controlled environment and in situ which conditions the evaluation in an application context linked to the stations. Thus, the laboratory tests allowed to quantify the capacity of the device on the range and the geometrical forms related to the equipment often present in stations, the tests on the effect of the surface according to the light on the basis of an experimental plan also allowed to study this effect of reflectance while taking into account the other important factors which intervene in the perturbation of the data of the device.

 The results obtained from this evaluation process provided recommendations and limits for the use of the LiDAR smartphone in a railway station environment (maximum surface area and volume to be scanned, identification of areas and objects at risk when using the tool, recommendations on the mode of movement during scanning, mode of use, etc.)

(Page 22, line 13-47)

Reviewer 2 Report

This paper presents analyses of the iPhone 12Pro LiDAR in the environment of railway station interiors through geometric evaluation of the indoor data on acquisition distance, reflection, and time. The results were compared with those from other scanners in a test environment similar to a railway station. Recommendations for the intervention of LiDAR for numerization in certain categories of railway station spaces were also provided. I think the paper cam be published with the following minor revision:

- 1st paragraph of “2 State of art”, “Some of his relevant work is presented …” (check “his”).

-        2nd paragraph of “2 State of art”, “…carried out by means of 3D BIM muck up…” (change “muck up” to “mock-up”).

-        Table 3, the precision of iPhone 12 Pro max LiDAR is “5 m - 10 m”, should “m” be “mm” or "cm"? Also, “Iphone” should be changed to “iPhone”.

Author Response

sensors-1958684

Title:

 Smartphone LiDAR Data: A Case Study for Numerisation Of Indoor Buildings In Railways’ Stations.

The combination of LiDAR with other technologies for numerisation is increasingly applied in the field of building, design, or geoscience, as it often brings time and cost advantages in 3D data survey processes. In this paper the reconstruction of 3D point cloud datasets is studied by an experimental protocol evaluation of new LiDAR sensors on smartphones. To evaluate and analyse the 3D point cloud datasets, different experimental trials are considered depending on the acquisition mode and the type of object or surface being scanned. The conditions allowing to obtain the most accurate data are identified to propose which acquisition protocol that seems to be the most adapted when using these LiDAR sensors to digitise complex interiors building such as railway stations. The aim of this paper is to propose: (i) a methodology to suggest the adaptation of an experiment protocol based on factors (distance, luminosity, surface, time , incidence) to assess the precision and accuracy of the smartphone LiDAR sensor in a controlled environment; (ii) a comparison, both qualitative and quantitative, of Smartphone LiDAR data with other traditional 3D scanner alternatives (Faro X130, VLX, Vz400i) while considering 3 representative building interior environments; and (iii) a discussion of the results obtained in a controlled and a field environment , making it possible to propose recommendations for the use of the LiDAR smartphone at the end of the numerisation  of the interior space of a building.

Comments for the reviewers:

First, we would like to thank you for your relevant comments and suggestions.

Please find a table listing our responses (in red) to comments (in black). Each response is structured in two steps:

- A detailed response to the reviewer’s comment in a small paragraph.

- A short summary of changes made to the paper with a reference to the page and line number.

Please find attached an updated version of our paper. Alterations resulting from the reviewer 1 (resp. reviewer 2, resp. reviewer 3) comments are highlighted in yellow (resp. green, resp. blue).

We hope this new version will meet your expectations.

Yours faithfully,

The authors.

Response to reviewer 2:

This paper presents analyses of the iPhone 12Pro LiDAR in the environment of railway station interiors through geometric evaluation of the indoor data on acquisition distance, reflection, and time. The results were compared with those from other scanners in a test environment similar to a railway station. Recommendations for the intervention of LiDAR for numerization in certain categories of railway station spaces were also provided. I think the paper cam be published with the following minor revision:

Overall response:

The authors thank the reviewers for their comments and interest in our work. We are pleased that our paper allows you to grasp the importance of our findings in the use of these new technologies in railways station spaces. We are glad that the topic of our paper is well understood and considered as interesting by the reviewer

We hope that our responses to the reviewer's comments meet their expectations and clarify our work.

Reviewer #2

Reviewer’s comments 

Responses and alterations made to the paper

Point 1: 1st paragraph of “2 State of art”, “Some of his relevant work is presented …” (check “his”).

Response 1: Thank you for this remark; we have added the references when they were missing

(Page2, line36-37)

Point 2: 2nd paragraph of “2 State of art”, “…carried out by means of 3D BIM muck up…” (change “muck up” to “mock-up”).

Response 2: Thank you for this remark; we have changed the word.

(Page2, line43)

Point 3: Table 3, the precision of iPhone 12 Pro max LiDAR is “5 m - 10 m”, should “m” be “mm” or "cm"? Also, “Iphone” should be changed to “iPhone”.

Response 3:  Thank you for this remark, we have put a correction of the values from 5m-10m to 5mm-20mm

(Page17, line25)

Reviewer 3 Report

The paper evaluates the use of Iphone12 ProMax lidar in a controlled test case and in a real case, and compares it with well known TLS equipment. The motivation for the work is clear. The RW is extensive, the methodology is well explained and the tests are thorough. This kind of work is relevant to know the capabilities of new devices.

However, there are some serious issues that need to be addressed:

1- The authors mention that the case study is a train station, but this is not very relevant since there are no characteristic elements of it and the studied areas correspond to any indooor area (parking (without cars), storage and rest area). A typical case study of a station should be added or the study should be changed to a generalization of indoor.

2- The recommendations of use that are concluded from this work are already well known. They should be related (numerically for example) to the tests performed.

3- > What software did the authors use? In [*], the authors talk about iPhone apps involve different behaviors, especially in overlapping points and gaps.

[*] Apple LiDAR Sensor for 3D Surveying: Tests and Results in the Cultural Heritage Domain

Minor issues:

Some punctuation marks have a space before. 

Where "outdoor spaces use" is not recommended, the authors may mean "open spaces", because this involes very open spaces even in indoor environment. 

indoor Mobile Mapping Systems instead of "Mobil scan with LiDAR (iMMs)".

Figure 5, why did authors only scan up to 3 meters, theoretically the iPhone reaches 5m? 

Another recent paper comparing Apple and Faro: New Trends in Laser Scanning for Cultural Heritage

Author Response

Manuscript ID

 sensors-1958684

Title:

 Smartphone LiDAR Data: A Case Study for Numerisation Of Indoor Buildings In Railways’ Stations.

The combination of LiDAR with other technologies for numerisation is increasingly applied in the field of building, design, or geoscience, as it often brings time and cost advantages in 3D data survey processes. In this paper the reconstruction of 3D point cloud datasets is studied by an experimental protocol evaluation of new LiDAR sensors on smartphones. To evaluate and analyse the 3D point cloud datasets, different experimental trials are considered depending on the acquisition mode and the type of object or surface being scanned. The conditions allowing to obtain the most accurate data are identified to propose which acquisition protocol that seems to be the most adapted when using these LiDAR sensors to digitise complex interiors building such as railway stations. The aim of this paper is to propose: (i) a methodology to suggest the adaptation of an experiment protocol based on factors (distance, luminosity, surface, time , incidence) to assess the precision and accuracy of the smartphone LiDAR sensor in a controlled environment; (ii) a comparison, both qualitative and quantitative, of Smartphone LiDAR data with other traditional 3D scanner alternatives (Faro X130, VLX, Vz400i) while considering 3 representative building interior environments; and (iii) a discussion of the results obtained in a controlled and a field environment , making it possible to propose recommendations for the use of the LiDAR smartphone at the end of the numerisation  of the interior space of a building.

Comments for the reviewers:

First, we would like to thank you for your relevant comments and suggestions.

Please find a table listing our responses (in red) to comments (in black). Each response is structured in two steps:

- A detailed response to the reviewer’s comment in a small paragraph.

- A short summary of changes made to the paper with a reference to the page and line number.

Please find attached an updated version of our paper. Alterations resulting from the reviewer 1 (resp. reviewer 2, resp. reviewer 3) comments are highlighted in yellow (resp. green, resp. blue).

We hope this new version will meet your expectations.

Yours faithfully,

The authors.

Response to reviewer 3:

The paper evaluates the use of Iphone12 ProMax lidar in a controlled test case and in a real case and compares it with well-known TLS equipment. The motivation for the work is clear. The RW is extensive, the methodology is well explained, and the tests are thorough. This kind of work is relevant to know the capabilities of new devices.

Overall response:

The authors thank the reviewers for his/her analyses of the potential publication of our work. We are pleased that our document is considered relevant for a better knowledge of the new devices.

We hope that our responses to the reviewer’s comments meet his/her expectations and clarify our work.

Reviewer #3

Reviewer’s comments 

Responses and alterations made to the paper

Point 1:  The authors mention that the case study is a train station, but this is not very relevant since there are no characteristic elements of it and the studied areas correspond to any indoor area (parking (without cars), storage and rest area). A typical case study of a station should be added, or the study should be changed to a generalization of indoor.

Response 1: Indeed, we are in the context of stations which are rather complex and varied buildings. the interiors of the stations are constituted at the same time of great spaces like the platforms of trains, the spaces travellers of the long corridors but also of the offices, rest area, sanitary spaces, warehouses, or rooms of technical buildings....

(Page6, ligne 1-28)

Page 7 , line5-12))

We have chosen these three environments based on criteria related to characteristics linked to digitisation constraints (surface area, space, requirement luminosity) and others in relation to the equipment which is intended to be digitised by the industrialist (Equipment : interior sinks, pipes, lamps, lockers, dispensers, dustbins, etc.).

The empty car park was chosen to test the limits of the devices on large surfaces and show the interior luminosity impact the point cloud data.

The storage with its cluttered environment characteristics corresponds to the railway station areas and allows to evaluate the acquisition capacity in areas overloaded with equipment.

The rest area has been tested because of its small surface and the equipment it contains, which are real cases of equipment present in stations and which are potentially digitizable in our industrial context.

A study on platforms or passenger areas (which are the most known areas in stations) could not be tested because of the low acquisition range that we observed on the iPhone during the tests in controlled environment. The use of the sensor was only limited to spaces not occupying large dimensions (9m² to 70m²).

(Page18, line16-23)

Point 2: The recommendations of use that are concluded from this work are already well known. They should be related (numerically for example) to the tests performed.

Response 2: Thank you for this remark, we have put a correction with numerical information in recommendation

(Page22, line15-42)

Page23-16-22

Point 3: What software did the authors use? In [*], the authors talk about iPhone apps involve different behaviours, especially in overlapping points and gaps.

[*] Apple LiDAR Sensor for 3D Surveying: Tests and Results in the Cultural Heritage Domain

Response 3: We use the software (iOS application) 3Dscan LiDAR. We have justified the choice of this software in the paragraph ''3.1. LiDAR Smartphone: IPhone12 Pro". Based on other research work and depending on the surfaces we scan; 3D scan had a better surface coverage capacity, a less noisy surface reconstruction and better export simplicity in point cloud format.

(Page 6, line14-16)

Point 4:  Some punctuation marks have a space before

Response 4: Thank you for this remark, we have put a correction

Point 5:   Where "outdoor spaces use" is not recommended, the authors may mean "open spaces", because this involves very open spaces even in indoor environment.

Response 5: Thank you for this remark, we have put a correction in recommendation.

(Page5, line12 Page22, line27)

Indeed, the term outdor environment can also be used by other authors in their work to refer to open spaces

(Page24, line34 Page25, line10)

Point 6:    Indoor Mobile Mapping Systems instead of "Mobil scan with LiDAR (iMMs)".

Response 6: Response 2: Thank you for this remark; we have changed

(Page4, line37-38)

Point 7: Figure 5, why did authors only scan up to 3 meters, theoretically the iPhone reaches 5m?

Response 7: Indeed, the IPhone LiDAR   range reaches 5 m, but we have observed that from the moment over 2.5 m, the observed noise standard deviation had a high variation and value which corresponds to the limits envisaged for digitisation in our context. Then we limited all our test to 3m range. The relatively stable standard deviation values observed below 2.5 m allowed us to carry out in SITU surveys not exceeding 2m to avoid significant errors in the accuracy Thank you for this remark; we have added the references.

(Page 8, ligne35-40, Page9 line,1)

Point 8: Another recent paper comparing Appel and FARO: New Trends in Laser Scanning for Cultural Heritage.

Response 8: Indeed, following your suggestion, we have the work done by who are quite close to the approach on the external facades. We have however integrated them in the bibliography to justify our choice of comparison with a traditional static scanner. However, in our approach we had to justify the choice of the FARO by comparing it with other types of scanners (VLX, Navis) before deciding on its choice for our comparison cloud to cloud.

Page 5-18-28

Reviewer 4 Report

The manuscript investigated the spatial mapping capability of the iPhone 12 Pro LiDAR in the environment of railway station interiors. The manuscript is easy to follow. I have only some minor comments:

(1).  page 7, line 15 and line 21: "Uses-case A and B correspond for the most part to maintenance in small spaces" and "Uses-case A and B correspond for the most part to maintenance in large spaces."  there is a contradiction in these two sentences.

(2) page 18, line 27: ICP is an algorithm for point cloud registration. I don't think it's a good choice to calculates the cloud-to-cloud deviations.

(3) page 20, line 45: "Cloud-to-cloud deviation measurements (LiDAR smartphone VS FARO) have identified risky areas to" the sentence is not complete, please check.

Author Response

Manuscript ID

 sensors-1958684

Title:

 Smartphone LiDAR Data: A Case Study for Numerisation Of Indoor Buildings In Railways’ Stations.

The combination of LiDAR with other technologies for numerisation is increasingly applied in the field of building, design, or geoscience, as it often brings time and cost advantages in 3D data survey processes. In this paper the reconstruction of 3D point cloud datasets is studied by an experimental protocol evaluation of new LiDAR sensors on smartphones. To evaluate and analyse the 3D point cloud datasets, different experimental trials are considered depending on the acquisition mode and the type of object or surface being scanned. The conditions allowing to obtain the most accurate data are identified to propose which acquisition protocol that seems to be the most adapted when using these LiDAR sensors to digitise complex interiors building such as railway stations. The aim of this paper is to propose: (i) a methodology to suggest the adaptation of an experiment protocol based on factors (distance, luminosity, surface, time , incidence) to assess the precision and accuracy of the smartphone LiDAR sensor in a controlled environment; (ii) a comparison, both qualitative and quantitative, of Smartphone LiDAR data with other traditional 3D scanner alternatives (Faro X130, VLX, Vz400i) while considering 3 representative building interior environments; and (iii) a discussion of the results obtained in a controlled and a field environment , making it possible to propose recommendations for the use of the LiDAR smartphone at the end of the numerisation  of the interior space of a building.

Comments for the reviewers:

First, we would like to thank you for your relevant comments and suggestions.

Please find a table listing our responses (in red) to comments (in black). Each response is structured in two steps:

- A detailed response to the reviewer’s comment in a small paragraph.

- A short summary ( of changes made to the paper with a reference to the page and line number.

- Some improvements on form, writing and clarity on some paragraphs are also made to the paper

Please find attached an updated version of our paper. Alterations resulting from reviewer 3 (resp. reviewer 4, resp.) comments are highlighted in. blue (resp. red).

We hope this new version will meet your expectations.

Comment in yellow (resp. green). Correspond to the first reviewing of the paper  for the reviewer 1 (resp. reviewer 2, resp. )

Yours faithfully,

The authors.

Response to reviewer 3:

The authors have answered the questions correctly and have made numerous changes accordingly. I congratulate the authors for their work and the results shown that contribute to clarify little by little the functioning of Apple's LiDAR.

Overall response:

The authors thank the reviewers for their comments. We are pleased that the responses and changes to our article will help you to clarify and better understand the importance of our findings in the use of these new technologies in railway station spaces.

Response to reviewer 4:

The manuscript investigated the spatial mapping capability of the iPhone 12 Pro LiDAR in the environment of railway station interiors. The manuscript is easy to follow. I have only some minor comments:

Overall response:

The authors thank the reviewers for their comments and interest in our work We are pleased that the subject of our article is well understood and easy to follow by the reviewer

We hope that our responses to the reviewer’s comments meet his/her expectations and clarify our work.

Reviewer #4

Reviewer’s comments 

Responses and alterations made to the paper

Point 1: page 7, line 15 and line 21: "Uses-case A and B correspond for the most part to maintenance in small spaces" and "Uses-case A and B correspond for the most part to maintenance in large spaces."  there is a contradiction in these two sentences.

Response 1: Thank you for this remark; we have corrected the sentence. We put “use-case  C and D” which corresponding  the most part to maintenance in large space’

(Page7, line13-20)

Point 2: (2) page 18, line 27: ICP is an algorithm for point cloud registration. I don't think it's a good choice to calculates the cloud-to-cloud deviations

Response 2: Thank you for this remark; Inn fact

The ICP algorithm was used to facilitate the alignment between the two-point clouds. A C2C function in CloudCompare was used to calculate the deviations between the two-point clouds. We have put a correction on the sentence .

(Page13, line 27-30)

Point 3: page 20, line 45: "Cloud-to-cloud deviation measurements (LiDAR smartphone VS FARO) have identified risky areas to" the sentence is not complete, please check.

Response 3:  Thank you for this remark, the sentence is completed.

(Page27, line 11-14)

Round 2

Reviewer 3 Report

The authors have answered the questions correctly and have made numerous changes accordingly. I congratulate the authors for their work and the results shown that contribute to clarify little by little the functioning of Apple's LiDAR.

Author Response

Manuscript ID

 sensors-1958684

Title:

 Smartphone LiDAR Data: A Case Study for Numerisation Of Indoor Buildings In Railways’ Stations.

The combination of LiDAR with other technologies for numerisation is increasingly applied in the field of building, design, or geoscience, as it often brings time and cost advantages in 3D data survey processes. In this paper the reconstruction of 3D point cloud datasets is studied by an experimental protocol evaluation of new LiDAR sensors on smartphones. To evaluate and analyse the 3D point cloud datasets, different experimental trials are considered depending on the acquisition mode and the type of object or surface being scanned. The conditions allowing to obtain the most accurate data are identified to propose which acquisition protocol that seems to be the most adapted when using these LiDAR sensors to digitise complex interiors building such as railway stations. The aim of this paper is to propose: (i) a methodology to suggest the adaptation of an experiment protocol based on factors (distance, luminosity, surface, time , incidence) to assess the precision and accuracy of the smartphone LiDAR sensor in a controlled environment; (ii) a comparison, both qualitative and quantitative, of Smartphone LiDAR data with other traditional 3D scanner alternatives (Faro X130, VLX, Vz400i) while considering 3 representative building interior environments; and (iii) a discussion of the results obtained in a controlled and a field environment , making it possible to propose recommendations for the use of the LiDAR smartphone at the end of the numerisation  of the interior space of a building.

Comments for the reviewers:

First, we would like to thank you for your relevant comments and suggestions.

Please find a table listing our responses (in red) to comments (in black). Each response is structured in two steps:

- A detailed response to the reviewer’s comment in a small paragraph.

- A short summary ( of changes made to the paper with a reference to the page and line number.

- Some improvements on form, writing and clarity on some paragraphs are also made to the paper

Please find attached an updated version of our paper. Alterations resulting from reviewer 3 (resp. reviewer 4, resp.) comments are highlighted in. blue (resp. red).

We hope this new version will meet your expectations.

Comment in yellow (resp. green). Correspond to the first reviewing of the paper  for the reviewer 1 (resp. reviewer 2, resp. )

Yours faithfully,

The authors.

Response to reviewer 3:

The authors have answered the questions correctly and have made numerous changes accordingly. I congratulate the authors for their work and the results shown that contribute to clarify little by little the functioning of Apple's LiDAR.

Overall response:

The authors thank the reviewers for their comments. We are pleased that the responses and changes to our article will help you to clarify and better understand the importance of our findings in the use of these new technologies in railway station spaces.

Response to reviewer 4:

The manuscript investigated the spatial mapping capability of the iPhone 12 Pro LiDAR in the environment of railway station interiors. The manuscript is easy to follow. I have only some minor comments:

Overall response:

The authors thank the reviewers for their comments and interest in our work We are pleased that the subject of our article is well understood and easy to follow by the reviewer

We hope that our responses to the reviewer’s comments meet his/her expectations and clarify our work.

Reviewer #4

Reviewer’s comments 

Responses and alterations made to the paper

Point 1: page 7, line 15 and line 21: "Uses-case A and B correspond for the most part to maintenance in small spaces" and "Uses-case A and B correspond for the most part to maintenance in large spaces."  there is a contradiction in these two sentences.

Response 1: Thank you for this remark; we have corrected the sentence. We put “use-case  C and D” which corresponding  the most part to maintenance in large space’

(Page7, line13-20)

Point 2: (2) page 18, line 27: ICP is an algorithm for point cloud registration. I don't think it's a good choice to calculates the cloud-to-cloud deviations

Response 2: Thank you for this remark; Inn fact

The ICP algorithm was used to facilitate the alignment between the two-point clouds. A C2C function in CloudCompare was used to calculate the deviations between the two-point clouds. We have put a correction on the sentence .

(Page13, line 27-30)

Point 3: page 20, line 45: "Cloud-to-cloud deviation measurements (LiDAR smartphone VS FARO) have identified risky areas to" the sentence is not complete, please check.

Response 3:  Thank you for this remark, the sentence is completed.

(Page23, line 11-14)
